# SIRLUT: Simulated Infrared Fusion Guided Image-adaptive 3D Lookup Tables for Lightweight Image Enhancement

Kaijiang Li
School of Computer and Artificial
Intelligence
Zhengzhou University
Zhengzhou, Henan, China
riversky@gs.zzu.edu.cn

Hao Li
School of Computer and Artificial
Intelligence
Zhengzhou University
Zhengzhou, Henan, China
lhlxr@gs.zzu.edu.cn

Haining Li
International College of Zhengzhou
University
Zhengzhou University
Zhengzhou, Henan, China
lihny@stu.zzu.edu.cn

Peisen Wang
School of Computer and Artificial
Intelligence
Zhengzhou University
Zhengzhou, Henan, China
wps28501@gs.zzu.edu.cn

Chunyi Guo*
School of Computer and Artificial
Intelligence
Zhengzhou University
Zhengzhou, Henan, China
gcy@zzu.edu.cn

Wenfeng Jiang
China Mobile Group Henan Company
Limited
Zhengzhou, Henan, China
13837189520@139.com

## Abstract

Researchers have applied 3D Lookup Tables (LUTs) in cameras, offering new possibilities for enhancing image quality and achieving various tonal effects. However, these approaches often overlook the non-uniformity of color distribution in the original images, which limits the performance of learnable LUTs. To address this issue, we introduce a lightweight end-to-end image enhancement method called Simulated Infrared Fusion Guided Image-adaptive 3D Lookup Tables (SIRLUT). SIRLUT enhances the adaptability of 3D LUTs by reorganizing the color distribution of images through the integration of simulated infrared imagery. Specifically, SIRLUT consists of an efficient Simulated Infrared Fusion (SIF) module and a Simulated Infrared Guided (SIG) refinement module. The SIF module leverages a cross-modal channel attention mechanism to perceive global information and generate dynamic 3D LUTs, while the SIG refinement module blends simulated infrared images to match image consistency features from both structural and color aspects, achieving local feature fusion. Experimental results demonstrate that SIRLUT outperforms state-of-the-art methods on different tasks by up to 0.88 ~ 2.25dB while reducing the number of parameters. Code is available at https://github.com/riversky2025/SIRLUT.

## CCS Concepts

• **Computing methodologies → Computer vision problems**; **Image processing**.

*Corresponding author.

## Keywords

Image enhancement, Photo retouching, 3D lookup tables, Simulated infrared fusion

### ACM Reference Format:
Kaijiang Li, Hao Li, Haining Li, Peisen Wang, Chunyi Guo, and Wenfeng Jiang. 2024. SIRLUT: Simulated Infrared Fusion Guided Image-adaptive 3D Lookup Tables for Lightweight Image Enhancement. In *Proceedings of the 32nd ACM International Conference on Multimedia (MM '24), October 28–November 1, 2024, Melbourne, VIC, Australia.* ACM, New York, NY, USA, 9 pages. https://doi.org/10.1145/3664647.3680918

## 1 Introduction

When taking photos with a camera, overexposure, underexposure, backlit conditions, and other shooting conditions may cause loss of highlight details, unclear shadow details, color distortion, or dark subjects. With the rapid development of deep learning, many algorithms [11–13, 24] based on deep neural networks have been applied to image enhancement tasks, particularly in addressing these issues, achieving satisfactory performance. Compared to traditional methods [1, 18, 27], deep learning-based approaches improve performance while simplifying the processing pipeline. However, these methods utilize complex network inference for dense pixel mapping, which not only consumes high computational and storage resources but also fails to meet lightweight requirements and ensure efficient operations on camera hardware.

Researchers utilize static 3D LUTs for color mapping and simplifying the inference process. Based on the RGB color space and trilinear interpolation, 3D LUTs achieve more accurate color adjustments, effectively improving the quality and effectiveness of image enhancement [7, 15, 34, 36, 40, 41]. Zeng et al. [38] first propose learnable 3D LUTs, achieving image enhancement by generating adaptive 3D LUTs based on image features. However, the uneven pixel distribution in the original image reduces the pixel utilization rate of the learnable 3D LUTs during the sampling process. Zhang et al. [41] propose an efficient hash-based 3D LUTs method called HashLUT. Building on the concept of hashing, they further develop a fast and lightweight image enhancement network. Nevertheless,

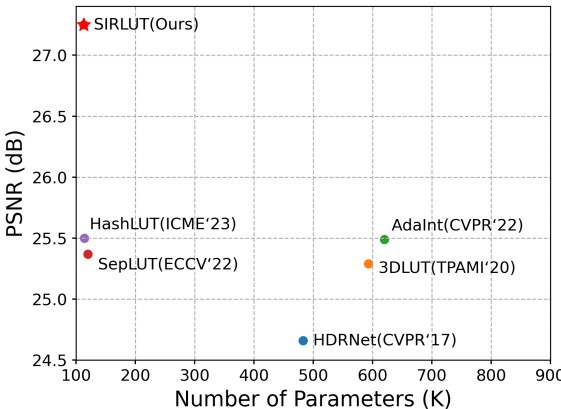

**Figure 1: PSNR results v.s the size of parameters of different methods for image enhancement on MIT-Adobe FiveK dataset[2]. Our method achieves the optimal results with the smallest parameters.**

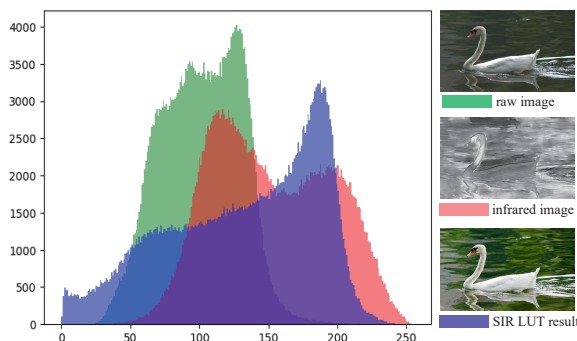

**Figure 2: The pixel distributions corresponding to the original, simulated infrared, and enhanced image are presented. The pixel values of the original image are mainly concentrated below 150. The simulated infrared image has two peaks in its pixel distribution, resulting in a more uniform distribution. The enhanced image leans towards a fusion of the original and simulated infrared image.**

the problem of limited feature extraction ability of the learnable 3D LUTs model due to uneven pixel distribution in original images has not been solved.

To balance the pixel value distributions of the original image, several methods increase feature components and adjust sampling methods to address the problem of uneven pixel distribution [5, 36, 37]. SepLUT [37] improves the utilization of 3D LUTs units by supplementing the ability of mixed color components with 1D LUTs separated from color components. AdaInt [36] improves the performance of 3D LUTs by introducing image adaptive sampling interval to learn non-uniform 3D LUTs layout. These methods achieve equalization of the pixel distribution of the original image features to a certain degree. The performance of previous state-of-the-art methods is partially shown in Figure 1.

In order to fundamentally solve the problem of uneven pixel distribution in the original images, we introduce the simulated infrared modality into the process of color mapping learning for adaptive 3D LUTs, aiming to obtain a more uniform pixel distribution. As shown in Figure 2, the pixel distribution of the original images is often concentrated due to the influence of the camera, which limits the effectiveness of the adaptive 3D LUTs learning ability and impairs network performance. In contrast, simulated infrared images reflect intensity information at the pixel level. Their pixel distribution is uniform, and their structural features can also serve as semantic-level features to enhance network performance. Specifically, we design a method called Simulated Infrared Fusion Guided Image-adaptive 3D Lookup Tables (SIRLUT). SIRLUT consists of a Simulated Infrared Fusion (SIF) dynamic 3D LUTs and a Simulated Infrared Guided (SIG) image refinement module. SIF utilizes a cross-modal channel attention mechanism to perceive global information and generate dynamic 3D LUTs. To address the problem of inconsistent local details in the image after enhancing with 3D LUTs, SIG refines the image from both structural and color perspectives using simulated infrared images. The combination of global adjustments and local refinement modules enables SIRLUT to achieve lightweight and efficient image enhancement.

The main contributions of this paper are as follows:

- We propose an end-to-end multimodal image enhancement network, SIRLUT, which utilizes simulated infrared modality to enhance the expressive power of 3D LUTs and improve the details in the enhanced images. To the best of our knowledge, this is the first work that employs simulated infrared images to supplement pixel distribution information in 3D LUT-based image enhancement.
- We propose the SIF and SIG modules. The SIF module achieves the extraction of global features through spatial channel transformation and cross-modal channel attention mechanism, while the SIG module conducts structural-consistent feature matching and local feature fusion for further image refinement.
- Experimental results on publicly benchmark datasets demonstrate the proposed image enhancement network significantly outperforms state-of-the-art image enhancement methods both quantitatively and qualitatively.

## 2 Related Work

### 2.1 Image Enhancement

Deep learning-based image enhancement networks can be divided into two categories. The first category refers to pixel-to-pixel image enhancement networks based on the U-shape architecture [29, 35, 42]. These methods have large computational overheads and are difficult to embed in devices such as cameras. The second category [14, 24] combines neural networks with physical methods, using color mapping models to achieve lightweight and fast image enhancement. Typical color mapping models include affine transformations [8, 32], curve-fitting functions [9, 17, 24], and 3D Lookup Tables [36–38]. In recent years, researchers have proposed various LUT-based image enhancement methods [4, 7, 15, 34, 36, 40, 41]. Zeng et al. [38] first propose a learnable 3D LUTs method, which utilizes multiple basic 3D LUTs for paired and unpaired learning

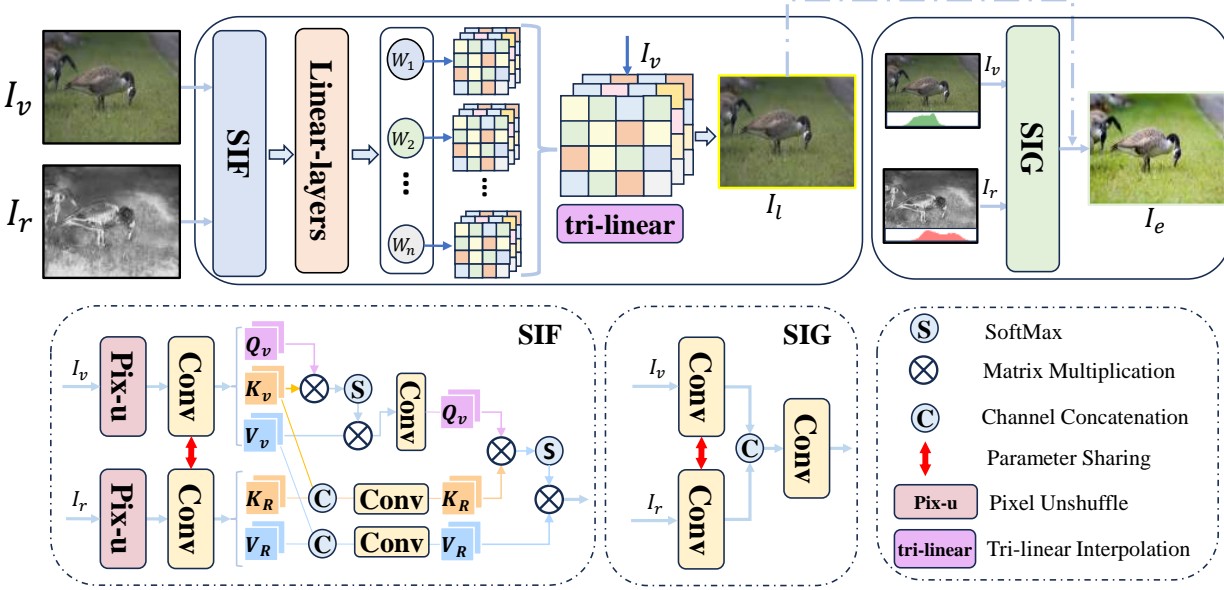

**Figure 3: The overview of SIRLUT. The SIRLUT consists of two parts: the Simulated Infrared Fusion (SIF) LUT and the Simulated Infrared Guided (SIG) detail optimization. The SIRLUT takes the original image $I_v$ and the simulated infrared image $I_r$ as inputs. It utilizes the SIF feature extraction model to generate n weight parameters $W_i$ and LUTs for dynamic mapping to enhance the image, resulting in the obtained enhanced image $I_l$. Finally, $I_l$ is refined using the SIG method to obtain the final image $I_e$.**

and uses a CNN to predict the weights of multiple 3D LUTs. Subsequently, these weights are merged into an adaptive 3D LUT for image enhancement. However, the uniform sampling strategy used in this method [38] limits the expressive power of 3D LUTs. AdaInt [36] performs dense sampling in highly nonlinear color ranges and sparse sampling in nearly linear color ranges, enhancing the expressive power of 3D LUTs. Some researchers also optimize the lightweight nature of 3D LUTs themselves. HashLUT [41] utilizes an efficient hash-based form to adaptively learn HashLUT, handling hash conflicts to solve the memory consumption issue of 3D LUTs in image enhancement. However, the parameter memory consumption inside 3D LUTs is not significant, and few studies focus on optimizing the weight prediction parameters of 3D LUTs. We significantly optimize the network for predicting weights by combining simulated infrared modality to enrich pixel distribution information and cross-modal channel attention.

## 2.2 Multimodal Image Processing

In recent years, multimodal fusion has received extensive research attention in the field of image processing [22, 23, 28, 43–45]. Cong et al. [5] achieve high-resolution image harmonization by introducing semantic foreground and background information and using their structural information to guide pixel-level optimization. However, the information provided by semantic segmentation modality is limited and cannot provide effective pixel distribution and texture features. Some scholars [19, 23] use more detailed features provided by infrared images to perform image processing tasks. These methods fuse visible light images with infrared images, retaining

the thermal radiation intensity features of infrared images and the fine details of visible light images, thereby producing clear high-brightness targets and rich image details. RTFNet [30] integrates visible light images and thermal infrared information, fully utilizing the advantages of thermal imaging cameras in various lighting conditions, and achieves accurate semantic segmentation tasks in complex urban scenes. Wang et al. [31] introduce a novel method to generate cross-modal paired images for RGB-IR Re-ID tasks, solving the differences between visible light images and infrared images and improving the accuracy of person re-identification. Therefore, infrared images are crucial for visual tasks. In image enhancement tasks, simulated infrared images have uniform pixel distribution and rich structural texture, eliminating noise introduced by strong infrared penetration compared to images captured by real infrared cameras. Our work introduces simulated infrared images to achieve a more efficient multi-modal 3D LUTs image enhancement method.

## 3 Method

### 3.1 Overall Architecture

As shown in Figure 3, SIRLUT consists of a learnable fusion of simulated infrared fusion 3D LUTs network and a simulated infrared-guided refinement. The former comprises a weight prediction backbone network for cross-modal fusion features and a trilinear interpolation color mapping module for 3D LUTs, which captures contextual information through spatial transformation and cross-modal channel attention mechanism. The latter consists of a multi-modal structure extraction and a refinement module for local multi-modal feature fusion synergy. In this section, we will discuss 3D LUTs

and trilinear interpolation, 3D LUTs with simulated infrared fusion, simulated infrared guided refinement, and loss function.

## 3.2 3D LUTs and Trilinear Interpolation

3D LUTs stand for 3D Lookup Tables, which are used in image processing and color correction to map input colors to output colors based on three-dimensional grids. 3D LUTs define a cube grid G consisting of $D^3$ points, where D represents the number of sampling points for each color channel, and the usual values in experiments are 1, 9, 17, 25, and 33. Each point in the grid G $\{P_{(x,y,z)}\}$ $x, y, z = 0, 1, ..., D-1$ defines the input RGB index values $\{r_{(x,y,z)}, g_{(x,y,z)}, b_{(x,y,z)}\}$ and the color-converted mapped values $\{r'_{(x,y,z)}, g'_{(x,y,z)}, b'_{(x,y,z)}\}$. Given the value of D, the RGB color space is uniformly discretized into the grid G, and different 3D LUTs have different color conversion RGB outputs, which is the learnable parameter of the proposed method, and when D=33, the number of parameters for a single 3D LUTs is 107.8k($3D^3$). Color mapping via 3D LUTs is actually achieved through two basic operations: positioning and trilinear interpolation. Given the input RGB color $\{r_{(i,j,k)}, g_{(i,j,k)}, b_{(i,j,k)}\}$, the first step is to index and locate the position coordinates $(i, j, k)$ of the RGB color in the grid G, The specific description is as follows:

$$i = \frac{r_{(i,j,k)}}{l}, j = \frac{g_{(i,j,k)}}{l}, k = \frac{b_{(i,j,k)}}{l}, \tag{1}$$

$$l = \frac{Color_{max}}{D}, \tag{2}$$

where $Color_{max}$ represents the maximum color value, and $l$ represents the sampling interval of the sampling grid G. After locating the input RGB color in grid G position, its adjacent 8 element points can be used to interpolate the color conversion output values $\{r'_{(i,j,k)}, g'_{(i,j,k)}, b'_{(i,j,k)}\}$ through trilinear interpolation. Assuming point P is located at a certain position in the RGB color grid G, the eight adjacent vertices of point P are $\{P_0, P_1, ..., P_7\}$. The formula for calculating trilinear interpolation is as follows:

$$
\begin{aligned}
C'_{(i,j,k)} &= (1-d_r)(1-d_g)(1-d_b)G(P_0) \\
&+ d_r(1-d_g)(1-d_b)G(P_1) \\
&+ d_r d_g(1-d_b)G(P_2) \\
&+ (1-d_r)d_g(1-d_b)G(P_3) \\
&+ (1-d_r)(1-d_g)d_b G(P_4) \\
&+ d_r(1-d_g)d_b G(P_5) \\
&+ d_r d_g d_b G(P_6) \\
&+ (1-d_r)d_g d_b G(P_7),
\end{aligned}
\tag{3}
$$

where $G(P_K), k \in [0, 7]$ represent the RGB color values of point $P_k$ in grid $G$. $d_r$, $d_g$, and $d_b$ are the corresponding channel interpolation weight coefficients, respectively, and $C'_{(i,j,k)}$ represents the color conversion mapping result.

## 3.3 3D LUTs with Simulated Infrared Fusion

Deep neural network blocks initially train the learnable simulated infrared fusion 3D LUTs on n-base 3D LUT blocks. Then, the dynamic LUT corresponding to the original image is rapidly inferred using n weights and n base LUTs. Finally, the dynamic LUTs are used to compute pixel values for the image on a per-pixel using trilinear interpolation.

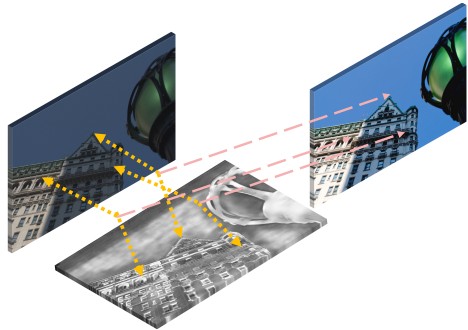

**Figure 4: The SIG refinement module achieves local feature fusion by combining simulated infrared images and matching image consistency features from both structure and color aspects. It is best viewed on a screen.**

**Weight prediction backbone network.**

To ensure the efficiency of the model, we propose a weight prediction backbone network, which incorporates a cross-modal channel attention mechanism to capture global information and generate dynamic 3D LUTs. This module allows the attention mechanism to operate at lower resolutions, effectively reducing computational complexity. We extract global features from the original image and simulated infrared image through three steps. First, we compress and transform image features using spatial-to-channel transformation. Next, we extract contextual features using the cross-modal channel attention mechanism. Finally, in the third step, we restore features using channel spatial transformation.

First, we align the resolutions of the original image $I_v$ and the simulated infrared image $I_r$ by utilizing the resize and spatial information rearrangement mechanism, resulting in the corresponding features $F_v$ and $F_r$. The specific description is as follows:

$$F = Unshuffle(\lambda, interpolate(I)), I \in \{I_v, I_r\}, \tag{4}$$

where $\lambda$ represents the conversion ratio used during the spatial transformation. Then, to reduce computational complexity and ensure the efficiency of the network feature extractor, we use a linear transformation with shared parameters to compress the number of features in $F_v$ and $F_r$ by a factor of $\mu$. The description is as follows:

$$F_v^e, F_r^e = Conv^*(\mu, F_v, F_r), \tag{5}$$

where $F_v^e$ and $F_r^e$ respectively represent the features after compression for $F_v$ and $F_r$. Then, we design a multi-head attention mechanism guided by simulated infrared for fused feature extraction. We perform a preceding channel expansion for $F_v^e$ and $F_r^e$ through linear transformations for multi-head attention. The specific procedure is described as follows:

$$
\begin{aligned}
\{Q_v, K_v, V_v\} &= Chunk(Conv(3m, F_v^e)), \\
\{K_r, V_r\} &= Chunk(Conv(2m, F_r^e)),
\end{aligned}
\tag{6}
$$

where $m$ represents the number of heads in the multi-head attention. We obtain the final query $Q_v$ by performing self-attention on the features of the original image, and self-attention is applied to the features of the simulated infrared image to obtain $K_r$ and $V_r$. Then, we conduct cross-channel attention feature extraction using $Q_v, K_r$,

and $V_r$. The calculation of attention is as follows:

$$Attention(Q, K, V) = Softmax(\frac{QK^T}{\xi})V, \qquad (7)$$

where $\xi$ is a learnable parameter. Then we extract features by sequentially applying self-attention to the original image and cross-modal channel attention. The cross-modal multi-head channel attention is described as follows:

$$\begin{aligned} Z &= \text{Concat}(Head^1, Head^2, ..., Head^m) \\ &= \text{Attention}(\text{Attention}(Q_v^m, K_v^m, V_v^m), K_r^m, V_r^m), \end{aligned} \qquad (8)$$

where $m$ represents the number of heads in the multi-head attention. Then, $Z$ is obtained by aggregating the parameters using adaptive average pooling, resulting in $F_{avgpool} \in \mathbb{R}^{C,2,2}$. Subsequently, a linear transformation with a stride of 2 is applied to convert the last two dimensions into one-dimensional features. Finally, a fully connected layer is used to transform the squeezed image into the weight matrix $W \in \mathbb{R}^n$ for the 3D LUTs.

$$W^n = Linear_n(Squeeze(Conv(Avgpool(Z)))), \qquad (9)$$

where $n$ represents the number of 3D LUTs base blocks. The length of the predicted weight vector $W$ is also $n$.

**LUT transformation.**

We define n learnable LUT base blocks $G_n \in \mathbb{R}^{D \times D \times D}$. The table lookup process for the LUT is described in Section 3.2. The LUT mapping for input image $I_l$ is as follows:

$$I_l = \sum_{i=1}^{n} W_i G_i(I_v), \qquad (10)$$

where $I_l$ represents the image after mapping conversion.

## 3.4 Simulated Infrared Guided Refinement

After the aforementioned processing by the SIF module, the enhanced image still fails to adequately capture local features. Therefore, we design the SIG module. As shown in Fig. 4, the SIG module achieves the capture of local features by extracting structurally consistent features between the simulated infrared image and the original image, as well as performing local feature fusion. The specific description is as follows:

We adopt shared parameter dilated convolution to learn the structural consistency between $I_v$ and $I_r$, and utilize the fused features to refine the pixels. The extraction of structural consistency between $I_v$ and $I_r$ is described as follows:

$$G_v, G_r = dilate\_conv^*(I_v, I_r), \qquad (11)$$

where $G_v$ and $G_r$ represent the features after structural alignment.

Then, we perform single-pixel convolution on the concatenated features of $G_v$ and $G_r$, followed by residual refinement with the activated and normalized $I_l$. Subsequently, we apply a single-pixel convolution with a kernel size of 1 to the fused features. Next, we apply the LeakyReLU activation function for non-linear processing of the outputs and normalize them using InstanceNorm2d. The specific description is as follows:

$$I_e = I_l + \Phi(Conv([G_v, G_r])), \qquad (12)$$

where $I_e$ represents the final enhanced image, and $\Phi$ denotes the cascaded use of LeakyReLU and InstanceNorm2d.

## 3.5 Loss Function

The overall framework can be trained end-to-end. Our training loss function is defined as follows:

$$\mathcal{L} = \mathcal{L}_{recon} + 0.0001 \times \mathcal{L}_s + 10 \times \mathcal{L}_m, \qquad (13)$$

where $\mathcal{L}_{recon}$ denotes the mean squared error (MSE) loss used as a reconstruction loss. According to [38], the smoothing term $\mathcal{L}_s$ and the monotonicity term $\mathcal{L}_m$ serve as regularization terms for the constraint of the look-up table (LUT).

## 4 Experiments

### 4.1 Datasets and Implementation

**Datasets for Photo Retouching and Tone Mapping.** We evaluate the effectiveness of SIRLUT on the publicly available MIT-Adobe FiveK dataset [2] and the PPR10K dataset [20]. To verify our proposed SIRLUT, we expand the dataset using existing simulated infrared image generation models [6]. The MIT-Adobe FiveK dataset [2] contains 5,000 raw format images, and we select the C version images from manually adjusted five authentic samples for training. Following convention [3, 37, 38], we divide the dataset into 4,500 pairs of training samples and 500 pairs of testing samples. The PPR10K dataset [20] contains 11,161 raw portrait photos, providing three versions of authentic samples (A/B/C), which are divided into 8,875 pairs of training samples and 2,286 pairs of testing samples for model validation [20, 36]. Our research involves photo retouching and tone mapping[39]. We utilize sRGB images and their corresponding simulated infrared images to evaluate the photo retouching task on the MIT-Adobe FiveK [2] and PPR10K datasets [20]. For the tone mapping task, we convert 16-bit CIE XYZ format images to 8-bit sRGB format and perform performance validation on the MIT-Adobe FiveK dataset [2]. We adopt evaluation standards previously used in research [38], including PSNR, SSIM, and the L2-distance in CIE LAB color space ($\triangle E_{ab}$). On PPR10K [20], we also include the human-centered evaluation metrics (denoted by the "HC" superscript).

**Implementation and Training Details.** We conduct experiments on the RTX 3080Ti graphics processing unit using the PyTorch environment [26]. For both datasets, we set the spatial transformation ratio $\lambda$ to 4 and the feature compression factor $\mu$ to 8. During training, we employ the same data augmentation settings as in [20], and we train the model with a batch size of 1. The training epochs for the MIT-Adobe FiveK and PPR10K datasets are set to 300 and 100 respectively. We use the Adam optimizer [16], set the initial learning rate to $2 \times 10^{-4}$, and gradually decrease it to $1 \times 10^{-5}$ using the cosine annealing strategy [21].

### 4.2 Comparison with State-of-the-Arts

We compare our method with state-of-the-art photo enhancement methods. To verify the efficiency of the SIRLUT, we select a large number of methods based on 3D LUTs. For a fair comparison of methods based on 3D LUTs, we set the $N_s$ of LUTs to 33 and the number of basic blocks $M$ to 3.

**Quantitative Comparison.** We quantitatively compare state-of-the-art methods using the PSNR, SSIM, and $\triangle E_{ab}$ evaluation metrics. Table 1 displays the comparison results for photo retouching task on the MIT-Adobe FiveK dataset[2]. RSFNet[25] adopts

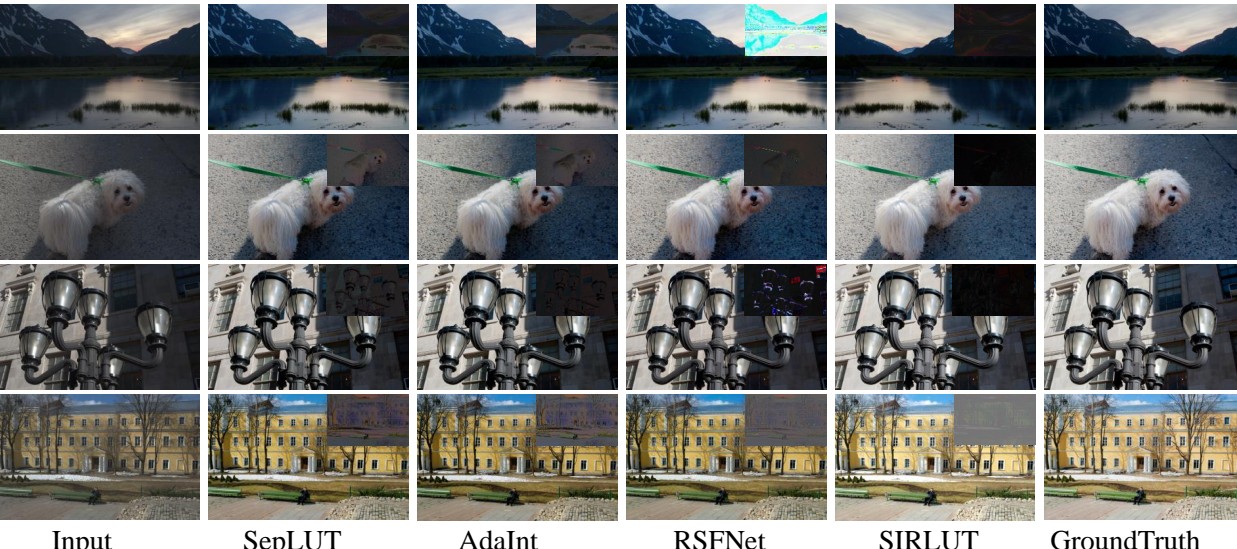

| Input | SepLUT | AdaInt | RSFNet | SIRLUT | GroundTruth |

**Figure 5: Qualitative comparison of photo retouching and error maps on the MIT-Adobe FiveK dataset [2]. Our method is obviously superior to other methods. The error maps are placed at the top-right of each image. Best viewed on screen.**

**Table 1: Quantitative comparisons on the MIT-Adobe FiveK dataset [2] for photo retouching. "/" means the results are absent in the original paper. The best performance is indicated by the color red, while the color blue indicates the second-best performance. Please note that the visualization is optimized for color viewing.**

| Method | Param. | PSNR↑ | SSIM↑ | $\triangle E_{ab} \downarrow$ |
|---|---|---|---|---|
| UPE[32] | 927.1K | 21.88 | 0.853 | 10.8 |
| DPE[3] | 3.4M | 23.75 | 0.908 | 9.34 |
| 3D LUT[38] | 593.5K | 25.29 | 0.923 | 7.55 |
| SA-3D LUT[33] | 4.5M | 25.50 | / | / |
| SepLUT[37] | 119.8K | 25.47 | 0.921 | 7.54 |
| HashLUT[41] | 114.0K | 25.50 | 0.926 | 7.46 |
| 3D-LUT+AdaInt[36] | 619.7K | 25.49 | 0.926 | 7.47 |
| SIRLUT | 113.3K | 27.25 | 0.942 | 6.19 |

**Table 2: Quantitative comparison on the PPR10K dataset [20] is conducted for portrait photo retouching, where a, b, and c represent the real data retouched by three experts. "/" indicates the results are absent in the original paper.**

| Method | E | PSNR↑ | $\triangle E_{ab} \downarrow$ | $\text{PSNR}^{HC} \uparrow$ | $\triangle E_{ab}^{HC} \downarrow$ |
|---|---|---|---|---|---|
| 3D-LUT[38] | a | 25.64 | 6.97 | 28.89 | 4.53 |
| 3D-LUT+HRP[20] | a | 25.99 | 6.76 | 28.29 | 4.38 |
| SepLUT[37] | a | 26.28 | 6.59 | / | / |
| HashLut[41] | a | 26.34 | 6.56 | / | / |
| 3D-LUT+AdaInt[36] | a | 26.33 | 6.56 | 29.57 | 4.26 |
| SIRLUT | a | 28.31 | 5.65 | 31.48 | 3.72 |
| 3D-LUT[38] | b | 24.70 | 7.71 | 27.99 | 4.99 |
| 3D-LUT+HRP[20] | b | 25.06 | 7.51 | 28.36 | 4.85 |
| SepLUT[37] | b | 25.23 | 7.49 | / | / |
| HashLut[41] | b | 25.42 | 7.40 | / | / |
| 3D-LUT+AdaInt[36] | b | 25.40 | 7.33 | 28.65 | 4.75 |
| SIRLUT | b | 27.67 | 5.89 | 30.82 | 3.84 |
| 3D-LUT[38] | c | 25.18 | 7.58 | 28.49 | 4.92 |
| 3D-LUT+HRP[20] | c | 25.46 | 7.43 | 28.80 | 4.82 |
| SepLUT[37] | c | 25.59 | 7.51 | / | / |
| HashLut[41] | c | 25.65 | 7.30 | / | / |
| 3D-LUT+AdaInt[36] | c | 25.68 | 7.31 | 28.93 | 4.76 |
| SIRLUT | c | 27.79 | 6.13 | 31.03 | 4.02 |

parallel region-specific filters, which may limit the model's ability to perform complex and fine adjustments to the image, resulting in poor image enhancement. SepLUT[37] improves the use of 3D LUTs, but in some cases, sub-transformations independent of components may lead to information loss or incompleteness, introducing errors. HashLUT[41] employs an approximation mapping method, which cannot accurately reflect the relationship between original data, and compared with traditional 3D LUTs, the nonlinear transformation effect is not accurate enough, resulting in limited enhancement effect. AdaInt[36] alleviates the problem of uneven pixel value distribution in the original image, but does not fundamentally supplement the additional pixel distribution information required. We supplement the pixel distribution information of the original image using simulated infrared image with uniformly distributed pixels and obvious semantic features. It is worth noting

that our method uses a shallower backbone network and optimizes the parameter quantity using SIF. On the MIT-Adobe FiveK dataset, our method achieves improvements of 1.75dB, 0.016, and 1.27 in terms of PSNR, SSIM, and $\triangle E_{ab}$ evaluation metrics respectively, compared to the state-of-the-art methods, while reducing the parameter count. As shown in Table 2 and 3, similar conclusions can be drawn regarding the comparison between photo retouching

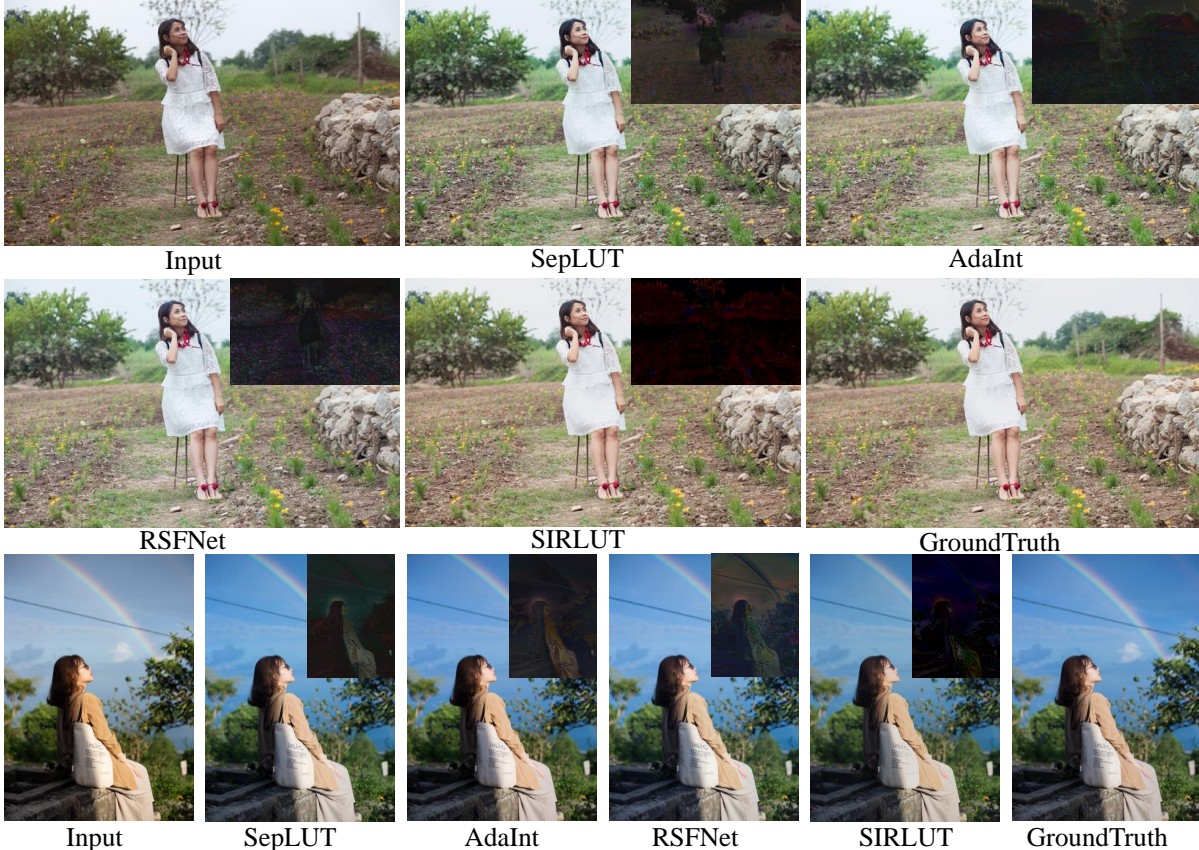

**Figure 6: Qualitative comparison of photo retouching and error maps conducted on the PPR10K dataset [20]. It can be found that the proposed method achieves better performance than other methods. The error maps are placed at the top-right of each image. It is best viewed on a screen.**

**Table 3: Quantitative comparison of tone mapping on the MIT-Adobe FiveK dataset [2]. The best performance is indicated by the color red, while the color blue indicates the second-best performance. Please note that the visualization is optimized for color viewing.**

| Method | PSNR↑ | SSIM↑ | $\triangle E_{ab}$ ↓ |
|---|---|---|---|
| UPE[32] | 21.56 | 0.837 | 12.29 |
| DPE[3] | 22.93 | 0.894 | 11.09 |
| HDRNet [8] | 24.52 | 0.915 | 8.04 |
| CSRNet[10] | 25.19 | 0.921 | 7.63 |
| 3D-LUT[38] | 25.07 | 0.920 | 7.55 |
| 3D-LUT+AdaInt[36] | 25.28 | 0.925 | 7.48 |
| SepLut[37] | 25.43 | 0.922 | 7.43 |
| SIRLUT | 26.31 | 0.932 | 6.74 |

**Table 4: Ablation analysis of SIF and SIG on the MIT-Adobe FiveK dataset [2].**

| SIF | SIG | PSNR↑ | SSIM ↑ | $\triangle E_{ab}$ ↓ |
|---|---|---|---|---|
| × | × | 24.85 | 0.914 | 7.94 |
| √ | × | 26.22 | 0.928 | 6.83 |
| × | √ | 25.82 | 0.926 | 6.93 |
| √ | √ | **27.25** | **0.942** | **6.19** |

datasets [20]. As shown in Figure 5, our method produces more satisfactory results in the photo retouching task on the MIT-Adobe FiveK dataset compared to other methods. Specifically, in the first row, The performance of RSFNet appears unstable under overexposed or low-light conditions due to the adoption of parallel region-specific filters. Our method effectively addresses the problem of underexposure. In the second row, while SepLUT and AdaInt improve the unit utilization rate of 3D LUTs by separating color components and non-uniform interval sampling, they fail to fundamentally solve the problem of uneven pixel distribution in original images, resulting in discontinuous transitions in smooth areas. Our method better handles image shadows caused by uneven illumination. In the third

task on the PPR10K dataset [20] and tone mapping task on the MIT-Adobe FiveK dataset[2].

**Qualitative Comparison.** We conduct a qualitative analysis of state-of-the-art methods on the MIT-Adobe FiveK [2] and PPR10K

row, RSFNet, SepLUT, and AdaInt exhibit poor handling of the reflection gloss of street lamps and building windows in close-up shots, demonstrating the superiority of our method in dealing with backlit shooting scenes. In the last row, the results obtained by the other methods are dark and lose many details, while our method preserves more texture details of buildings. As shown in Figure 6, the results on PPR10K demonstrate the efficiency of our method in handling portraits with background blurring and depth of field effects, and we observe that our method retains more background details, even when the background of the portrait is complex.

## 4.3 Ablation Studies

In this section, we conduct several ablation experiments on the photo retouching task on the MIT-Adobe FiveK dataset [2].

**Simulated infrared images.** To validate the effectiveness of simulated infrared images in SIRLUT, we gradually incorporate SIF and SIG into the 3D LUTs baseline on the MIT-Adobe FiveK dataset [2]. SIF refers to the fusion of simulated infrared modality during the LUT enhancement process, while SIG pertains to the introduction of simulated infrared modality during the refinement process. As shown in Table 4, the first row displays results without using simulated infrared modality, the second and third rows show results with the introduction of SIF and SIG modules respectively, while the last row displays the results obtained from the joint use of SIG and SIF. It is evident that the introduction of only the SIF module significantly enhances the model's ability to extract global features and handle image details. While using only the SIG refinement module also improves the model's performance, the effect is not as significant as the SIF module. However, due to the complementary nature of SIF and SIG, SIRLUT exhibits significant performance improvements in both color feature mapping 3D LUTs network and image refinement.

**Simulated infrared fusion strategy.** To evaluate the impact of the simulated infrared modality on the dynamic prediction weights of 3D LUTs, we conduct a comprehensive analysis of different fusion strategies on global feature extraction. We compare the influence of fusion methods based on linear transformation (CNN), spatial attention, and SIF on the model. Due to the small local receptive field of CNN, it cannot effectively capture global features. The spatial attention mechanism focuses on per-pixel feature extraction but often overlooks global context, resulting in high computational complexity and resource consumption. As shown in Table 5, it can be observed that SIF performs the best in both algorithmic model and enhancement effects, with the smallest number of parameters. This is because the simulated infrared modality as a complement to pixel value distribution information can greatly enhance the unit utilization rate of 3D LUTs. Additionally, the dynamic weight prediction guided by cross-modal channel attention avoids long-range dependencies of receptive fields and a large number of matrix operations in the global feature extraction process.

**Spatial transformation ratio and feature compression factor.** In the structure of our SIF module, we integrate a strategy that combines spatial dimension transformation with channel dimension compression. This approach aims to minimize the computational cost while improving the operational efficiency of the model. As shown in Figure 7, spatial transformation plays an important role

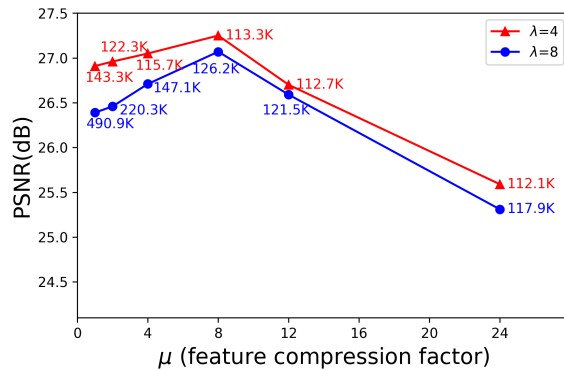

**Figure 7: Effects of spatial transformation ratio $\lambda$ and feature compression factor $\mu$ on the photo retouching performance on the MIT-Adobe FiveK dataset [2]. Note: The labels on the points indicate the size of the parameter of the models.**

**Table 5: Parameter quantity and effectiveness of different infrared fusion strategies in photo retouching on MIT-Adobe FiveK dataset[2].**

| Method | Param. | PSNR↑ | SSIM↑ | $\triangle E_{ab}$ ↓ |
|---|---|---|---|---|
| CNN | 115.1K | 26.30 | 0.928 | 6.80 |
| Space-att | 223.0K | 26.35 | 0.929 | 6.79 |
| SIF | 113.3K | **27.25** | **0.942** | **6.19** |

in adjusting the size of the model's parameters and improving its performance, compared to the adjustment factors of lambda ($\lambda$) and feature compression ($\mu$). When $\lambda$ is set to 8, an augmented spatial transformation ratio may impair the resolution of the channel attention mechanism in the context feature extraction process. Even with $\mu$ adjusted to 8, the performance does not reach the expected level. Nonetheless, setting $\lambda$ and $\mu$ to 4 and 8, respectively, allows the model to achieve an optimal balance between parameter efficiency and performance. Employing this configuration, our method attains remarkable outcomes, where the PSNR reaches 27.25dB, SSIM is 0.942, and the $\triangle E_{ab}$ value stands at 6.19.

## 5 Conclusions

In this paper, we propose a novel 3D LUT-based multi-modal image enhancement network called SIRLUT, which integrates simulated infrared fusion 3D LUTs and simulated infrared guided refinement to address the uneven pixel distribution in the original images. The SIF module achieves efficient 3D LUTs through spatial transformation and cross-modal channel attention mechanism. Meanwhile, the SIG module merges structurally consistent features from both simulated infrared images and the original images, incorporating local feature fusion for further refinement of the images. The combination of these two modules results in our method exhibiting exceptional performance and efficiency. Our proposed method not only achieves a significant visual enhancement effect but also is suitable for integration into both software and hardware devices such as cameras, mobile phones, and image processing software.

# Acknowledgments

This work is partially supported by the Joint Fund of the Ministry of Education for Equipment Pre-research (No. 8091B032257).

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
