# OpenReview forum: "SIRLUT: Simulated Infrared Fusion Guided Image-adaptive 3D Lookup Tables for Lightweight Image Enhancement"
_acmmm.org/ACMMM/2024/Conference — MM2024 Poster_

### Official Review · Reviewer_PCBx · 2024-05-21

**Rating:** 5
**Confidence:** 3

**Summary:**

This paper proposes a novel image enhancement method named Simulated Infrared Fusion Guided Image-adaptive 3D Lookup Tables (SIRLUT). The method addresses the challenges posed by non-uniform color distribution in images, which limits the performance of traditional 3D Lookup Tables (LUTs). SIRLUT introduces simulated infrared imagery to reorganize color distribution and enhance the adaptability of 3D LUTs. The method comprises two main components: the Simulated Infrared Fusion (SIF) module, which uses a cross-modal channel attention mechanism to generate dynamic 3D LUTs, and the Simulated Infrared Guided (SIG) refinement module, which blends simulated infrared images to achieve local feature fusion. The authors claim that SIRLUT outperforms state-of-the-art methods in various tasks, achieving significant performance improvements with fewer parameters.

**Strengths:**

1. The introduction of simulated infrared imagery to enhance 3D LUTs for image processing is innovative and addresses the problem of non-uniform color distribution effectively.
2. The paper is well-organized, with clear explanations of the methodology, experimental setup, and results.
3. Experimental results on publicly available datasets (MIT-Adobe FiveK and PPR10K) show that SIRLUT outperforms existing methods by a notable margin (up to 2.25dB in PSNR), while also reducing the parameter count, making it more efficient.

**Limitations:**

1. While the overall approach is novel, the individual components such as 3D LUTs and cross-modal attention mechanisms are well-established in the literature. The novelty primarily lies in their combination and application.
2. The presence of typos, Linner layers ->Linear layers in Figure 3.
3. Simulating infrared images is a key component of SIRLUT. This paper should provide some details on infrared image generation and simulate the impact of infrared images on network performance.
4. Although the method in the paper has a lower number of parameters, the running time of the network should also be provided to confirm the statement of efficiency and lightweight performance.
5. The paper lacks a description of the loss function.

**Suitability:**

3

---

### Official Review · Reviewer_gbDu · 2024-05-22

**Rating:** 4
**Confidence:** 3

**Summary:**

This paper proposes SIRLUT, a method utilizing a simulated infrared image fusion strategy and 3D lookup table for lightweight image enhancement. Specifically, the authors designed the Simulated Infrared Fusion (SIF) module and the Simulated Infrared Guided (SIG) refinement module to fully exploit the additional information from simulated infrared images, extracting consistent features in structure and color. The enhanced images exhibit a wider pixel distribution, presenting better visual effects and achieving significant improvements in quantitative metrics. The authors also conducted ablation studies to demonstrate the effectiveness of the proposed modules.

**Strengths:**

1. The paper innovatively proposes an image-adaptive 3D LUT generation method based on infrared image fusion and an infrared image-guided image reconstruction method. This method leverages information from simulated infrared images to complement the original visible light images, thereby enhancing image enhancement effects. In the simulated infrared image fusion module, the authors reasonably utilized the attention mechanism and provided detailed pipeline explanations.
2. The proposed new method achieved substantial improvements (up to 0.88~2.25dB) compared to the state-of-the-art methods, and the authors provided appropriate explanations for the principle of model performance improvement with simulated infrared images.

**Limitations:**

1. There may be a lack of discussion regarding the practical application scenarios of the proposed method. For instance, previous methods (such as 3D-LUT and SepLUT) have demonstrated near real-time model efficiency, whereas this paper lacks a discussion on the operational efficiency after introducing more complex network modules. Additionally, the authors mentioned that the simulated infrared images were synthesized using an infrared image synthesis method based on visible light images, implying that additional infrared images need to be introduced when applying this method. It is worth further discussing whether this limitation will affect the practical application of the proposed method.
2. The details of the paper may need further refinement. For example, in Figures 5 and 6, there are some smaller sub-images in the upper right corner of the images, and the authors did not seem to explain the meaning of these sub-images, which may affect the understanding of researchers unfamiliar with this research field. Additionally, there are some typographical errors in the paper, such as "Linner-laysers" in Figure 3, which seems to be "Linear-layers" based on the context, and the paragraph heading "Quatitative Comparison" around line 683, which should be "Quantitative Comparison." The authors are advised to carefully review the manuscript to correct these errors.

**Suitability:**

2

---

### Official Review · Reviewer_UTKz · 2024-05-24

**Rating:** 3
**Confidence:** 3

**Summary:**

- SIRLUT is a multimodal image enhancement network that utilizes simulated infrared modality to enhance 3D LUT. By leveraging pixel distribution information, it effectively extracts global features (via SIF) and improves spatial fidelity (via SIG), thereby enhancing the details in the processed images.

**Strengths:**

- By leveraging the pixel-level details information provided by the simulated infrared modality, SIRLUT enhances the perceptual quality of the images.

**Limitations:**

- The strength and weakness of the paper lie in the fact that the most significant performance improvement comes from using an infrared dataset generated by another method as an additional input for training. This contribution seems somewhat weak as a main contribution.
- It appears that the main improvement comes from the infrared image's uniformity compensating for the non-uniformity of the RAW data. However, the paper lacks an in-depth analysis of the distribution mismatch. I was curious about how the mismatch was utilized or overcome, but the paper merely states that SIG learned spatial fidelity. The role of pixel unshuffling, which might have had a normalizing effect, is not discussed at all.
- I spent quite some time trying to understand what "Linner-laysers" in Fig. 3 referred to. Both words were misspelled, making it hard to comprehend, which is critical for a figure explaining the main architecture.
- The reduction in parameters does not seem substantial enough to emphasize efficiency. It would be better to consider improvements in speed and memory efficiency as well.

**Suitability:**

3

---

### Official Review · Reviewer_FoSn · 2024-05-27

**Rating:** 3
**Confidence:** 3

**Summary:**

The paper introduces SIRLUT, an advanced method for image enhancement utilizing simulated infrared fusion and 3D Lookup Tables (LUTs). SIRLUT addresses the challenge of uneven pixel distribution in original images by integrating simulated infrared imagery to reorganize color distribution, enhancing the adaptability and performance of 3D LUTs. It consists of two main modules: the Simulated Infrared Fusion (SIF) module, which captures global information using cross-modal channel attention, and the Simulated Infrared Guided (SIG) refinement module, which improves local detail consistency. Experimental results demonstrate SIRLUT's superiority over state-of-the-art methods, achieving significant improvements in image quality with fewer parameters.

**Strengths:**

1. This paper pioneeringly proposes the use of simulated infrared images to supplement pixel distribution information in 3D LUT-based image enhancement.
2. The SIF and SIG modules proposed in this paper efficiently integrate information from simulated infrared and original RGB images, enabling SIRLUT to achieve excellent results with a smaller number of model parameters, thereby enhancing the feasibility of deployment on camera devices.
3. The ablation experiments and comparative experiments in this paper are comprehensive, validating the effectiveness of each module in the proposed model and demonstrating its superior performance.

**Limitations:**

1. Although the paper focuses on improving the performance of 3D LUT-based image enhancement methods by using models that integrate simulated infrared images, I believe that the comparative experiments should not be limited to 3D LUT-based methods. For example, other image enhancement methods utilizing simulated infrared images mentioned in the related work section, such as [19] and [23], should also be included.
2. Using a 3D LUT method that integrates simulated infrared images introduces an additional input from the simulated infrared image. Therefore, the time and space overhead required to obtain the simulated infrared image from the original image should not be overlooked. When comparing model parameters, inference latency, and memory consumption with other methods, the authors should comprehensively consider these factors.
3. A minor issue with this paper: What does the small image in the top right corner of each result image in Figures 5 and 6 represent? This small image covers the detail information of the original image. Could it be placed elsewhere? If it is not very important, it might be better to remove it.
4. Within the SIF module, it is recommended to conduct an ablation experiment: remove the self-attention mechanism for 𝐼𝑣 , allowing the upper branch to directly pass 𝐼𝑣  through Pix-U Block and Conv Block to obtain 𝑄𝑣, and then perform cross-attention with the lower branch. This experiment can help identify potentially redundant modules, thereby reducing the time and space complexity of the model's computation.


I think highly of your response and will consider increasing your rating based on your response.

**Suitability:**

3

---

### Meta-Review · Area_Chair_ix49 · 2024-07-03

**Recommendation:** Accept (Poster)
**Confidence:** 4

**Metareview:**

This paper received two borderline rejects, one borderline accept, and one weak accept. Reviewers FoSn, gbDu and PCBx all acknowledged the novelty of this work. Though the experiments are not sufficent and more methods should be compared, meta reviewer believes the novelty weighs more than extensive comparisons. Based on the innovation of this work, meta reviewer made the decision.